# Animal-Assisted Therapy Improves Communication and Mobility among Institutionalized People with Cognitive Impairment

**DOI:** 10.3390/ijerph17165899

**Published:** 2020-08-14

**Authors:** Maylos Rodrigo-Claverol, Belén Malla-Clua, Carme Marquilles-Bonet, Joaquim Sol, Júlia Jové-Naval, Meritxell Sole-Pujol, Marta Ortega-Bravo

**Affiliations:** 1Primary Health Care Center Bordeta-Magraners, Catalan Institute of Health, 25001 Lleida, Spain; bmalla.lleida.ics@gencat.cat (B.M.-C.); cmarquilles.lleida.ics@gencat.cat (C.M.-B.); jjove.lleida.ics@gencat.cat (J.J.-N.); 2Ilerkan Association, 25005 Lleida, Spain; merisole@gmail.com; 3Research Support Unit Lleida, Fundació Institut Universitari per a la recerca a l’Atenció Primària de Salut Jordi Gol i Gurina (IDIAPJGol), 08007 Barcelona, Spain; jsol.lleida.ics@gencat.cat (J.S.); mortega.lleida.ics@gencat.cat (M.O.-B.); 4Institut Català de la Salut, Atenció Primària, 25007 Lleida, Spain; 5Metabolic Physiopathology Research Group, Experimental Medicine Department, Lleida University-Lleida Biochemical Research Institute (UdL-IRBLleida), 25198 Lleida, Spain; 6Research Group in Therapies in Primary Care, Research Support Unit Lleida, Fundació Institut Universitari per a la recerca a l’Atenció Primària de Salut Jordi Gol i Gurina (IDIAPJGol), 08007 Barcelona, Spain

**Keywords:** aging, animal-assisted therapy, cognitive impairment, dementia, nursing homes, primary health care

## Abstract

Disorders of communication, social relationships, and psychomotricity are often characterized by cognitive impairment, which hinders daily activities and increases the risk of falls. This study aimed to evaluate the efficacy of an animal-assisted therapy (AAT) program in an institutionalized geriatric population with cognitive impairment. The variables evaluated included level of communication and changes in gait and/or balance. We performed a two-arm, parallel controlled, open-label, nonrandomized cluster clinical trial in two nursing home centers from an urban area. Patients in the two centers received 12 weekly sessions of physiotherapy, but the experimental group included AAT with a therapy dog. The study included a total of 46 patients (23 Control Group [CG], 23 Experimental Group [EG]) with a median age of 85.0 years. Of these, 32.6% had mild–moderate cognitive decline (Global Deterioration Scale of Reisberg [GDS] 2–4) and 67.4% severe cognitive decline (GDS 5–6). After the intervention, patients in the CG and EG showed a statistically significant improvement in all the response variables. When comparing both groups, no statistically significant differences were found in any of the Tinetti scale results (measuring gait and balance). However, the communication of patients in the EG, measured on the Holden scale, showed a statistically significant greater improvement postintervention than that of patients in the CG. AAT can be useful as a complementary, effective treatment for patients with different degrees of cognitive decline.

## 1. Introduction

Aging is a complex, progressive, and irreversible physiological process that involves biological, psychological, and social factors [1]. There are several pathologies associated with aging, including cognitive impairment, which can vary from mild to very severe. The advanced phases of cognitive decline (moderate-severe, severe, and very severe) accompanied by functional loss are known as dementia, which entails a global deterioration of cognitive functions, conduct disorders, mood, and sleep. Individuals suffering from dementia also present changes in mobility that interfere with their social and instrumental activities, and eventually with basic tasks of daily life [2,3].

Between 60% and 65% of institutionalized elderly people are estimated to have difficulties with stability and gait [4]. The prevalence of falls among this group is 50% [5]. Falls are considered to be one of the main causes of injury, disability, and even death in elderly patients [6], and they are a major public health problem [7]. One of the most useful methods for preventing falls is to work on improving balance, which is essential for correct mobility [4].

Another consequence of cognitive decline is the progressive impairment of the sufferers’ communication, which is a basic component of people’s functions and relationships [8]. To mitigate this problem, social stimulation is an important aspect of therapeutic activities in geriatric centers, and is designed to decrease social isolation, maintain or stimulate mental capacities, and increase awareness of the external environment [9]. In this context, and since currently available drugs have limited capacity to slow down the progression of dementia symptoms [10], it is necessary to develop nonpharmacological methods comprehensively treat people with cognitive impairment or dementia [11,12].

In addition to nonpharmacological treatments, animal-assisted therapy (AAT), included within animal-assisted interventions (AAI) together with animal-assisted activities and animal-assisted education [13], is widely used to enhance the results of such treatments [14]. AAI consists of the participation of animals in therapeutic or educational interventions to promote health or education and human well-being [15]. AAT sessions may have a relaxing effect on people with dementia and promote participant motivation by involving an animal as a therapeutic mediator [16]. One study found that patients with brain injuries receiving AAT focused on the same activity for a longer time and demonstrated greater enthusiasm for new exercises [17].

Regarding mobility, there is some emerging evidence that AAT may benefit patients with hemiparesis due to stroke [18,19,20], with aphasia [21], and with traumatic brain injury [22]. Walking with a dog can stimulate correct posture, regain momentum, and promote proper movement [23], as well as improve balance and gait function [22]. In addition, the interaction between the patient, the therapy animal, and the therapist has been shown to create a context that improves communication and confidence [24].

The overall assessment of the included studies indicates moderate effects of dog-assisted activities in cognitive disorders. However, the majority of studied outcome measures showed no significant effect [15]. In a current systematic review by Yakimicki et al. [25], the majority of the 32 studies included in the review highlight various benefits of utilizing AAI to alleviate some symptoms of dementia or to improve social behavior. The authors also point out some important limiting factors, making it difficult to compare studies, such as the lack of randomization, the absence of control groups, the lack of clearly defined study protocols and the methodological variability among animal-assisted therapy studies [26].

In most of these studies, validity is impeded by methodological weaknesses and low numbers of participants, and results need to be confirmed in more rigorous, larger scale studies. An important unanswered question is if the presence of the animal provides additional benefits (in comparison with interventions solely based on human interaction) [27].

For these reasons, the main objective of this study was to evaluate whether AAT can improve communication and mobility in an institutionalized geriatric population with cognitive impairment or dementia compared to a non-AAT intervention. This potential improvement was evaluated by assessing variation in the level of communication and in gait and/or balance.

## 2. Materials and Methods

### 2.1. Design and Participants

We performed a two-arm, parallel controlled, open-label, nonrandomized cluster clinical trial in two centers. The study sample included 110 institutionalized geriatric patients from two nursing homes pertaining to the same private entity in the Spanish city of Lleida. We assigned patients in center 1 (N = 40) as the experimental group (EG) and patients in center 2 (N = 70) as the control group (CG). We assigned the groups to independent geriatric centers because the presence of the dog could not be masked and bias could result if patients in the EG and CG shared the same center.

The inclusion criteria of our study were: institutionalized geriatric patients ≥65 years old, with a diagnosis of cognitive deterioration (Global Deterioration Scale of Reisberg (GDS) ≥2, ≤6) [28], and risk of falls according to Tinetti scale (≤24) [29]. After an initial interview, we excluded patients with very severe cognitive deterioration (GDS = 7), or who had allergies to or fear of animals. After applying these criteria, we excluded one subject who was <65 years old, three subjects that expressed fear of dogs, 25 subjects with severe cognitive impairment, and one subject who reported having a dog allergy. Our study included 81 participants, of which 25 belonged to center 1 and 56 to center 2. Finally, for the sake of equity, we randomly invited 25 of the 56 patients of center 2 to participate in the study. Thus, our final study sample consisted of 50 participants (center 1 (EG), N = 25; center 2 (CG), N = 25). Participants were informed about the study and were asked to sign an informed consent. In order to ensure comprehension by patients with severe cognitive impairment (GDS 5–6), consent was also asked to the caregiver with power of attorney.

A total of 50 participants were included, 25 in each of the two groups (CG and EG). The dropout rate was 8% (two participants) for each group, and the final sample consisted of 46 participants, 23 in the CG and 23 in the EG. The percentage of women in the final sample was 76.1%.

Table 1 describes the baseline characteristics and response variables in each group. The median age of the final sample was 85.0 (IQR 80.2; 87.0) years, with a similar distribution in the CG (86.0 (IQR 80.5; 87.5)) and EG (83.0 (IQR 80.5; 86.5)). The proportion of women was slightly higher in the CG (19, 82.6%) than in the EG (16, 69.6%). Regarding cognitive impairment, 4.35% of the total had very slight cognitive impairment (GDS 2), 13.0% mild cognitive impairment (GDS 3), 10.9% moderate cognitive impairment (GDS 4), 23.9% moderate-severe cognitive impairment (GDS 5), and 43.5% severe cognitive impairment (GDS 6). We found no statistically significant differences between groups for any of the above-mentioned variables (Table 1).

### 2.2. General Procedures

In the EG, we applied a physiotherapy and social stimulation program with the participation of the therapy dog (AAT), and in the CG we performed the same program without the animal. In the EG, we sought to promote physical contact with the animal (petting, brushing, throwing balls, feeding, or drinking, etc.), intending to promote communication and relationships between the participants and generate a casual and relaxed environment. The specifics of the sessions are described in Appendix A. The intervention was carried out at the facilities of both centers. We chose the nursing home center with the largest multipurpose room, where the AAT program was carried out, as the one hosting the EG and the other center as the one hosting the CG.

In both groups (EG and CG), we performed one 60 min session per week for 12 weeks. The sessions were carried out in small groups of six patients (four groups in EG and four in CG), in which different physiotherapy exercises were performed to improve gait and balance (Appendix A). The experiment was carried out in two periods of three months, and the eight groups were equally distributed during these two periods. All sessions included specific objectives and were previously designed and agreed on between the different professionals, with guidance from the physical therapist, psychologist, and occupational therapist at the nursing home. Six sessions focused on improving balance and six sessions on gait. In all sessions, exercises were carried out to enhance communication between group members. In each session, we encouraged the patients to talk about different topics (e.g., pets, traditional festivals, food, hobbies, news, seasons of the year, etc.).

### 2.3. Human and Animal Resources

The sessions were facilitated by the same two primary care nurses in both groups (EG and CG). The EG also included a family doctor with a track record in dog training and technical training in AAT since 2007. The family doctor was the coordinator of the project. She participated in the sessions as the AAT technician, and was the responsible of the dog, its training, care, and well-being, as well as the dog–patient interaction. She is part of the professional team of the Ilerkan association (www.ilerkan.com), which is a nonprofit association dedicated to AAI and has a civil liability insurance. The physiotherapist and the occupational therapist at the nursing home collaborated in planning the sessions.

We chose a dog as the AAT animal because of its great ability to interact with people and stimulate them. The intervention involved one therapy dog that conducted two sessions per week during the 6 months of the study. The dog was a five-year-old male German shepherd who was trained with positive training techniques using the Clicker method [30,31] and was selected based on his character, aptitude and training using the Liackhoff test [32]. He had been working regularly with his trainer in different centers, so he was used to interacting with strangers. He was regularly monitored by a veterinarian doctor and always worked with his trainer, the family doctor. The Ilerkan association internal protocols concerning health aspects and animal wellbeing are specified in Appendix A.

### 2.4. Measurements

To evaluate the effects of the intervention, we measured the response to the variables of interest by using standard scales applied in physiotherapy and nursing care.

We evaluated communication using the Holden Communication Scale, [33,34] which consists of 12 items for assessing communication, and is divided into three sections. The first entails observing parameters such initiative, interest, pleasure, and humor; the second evaluates knowledge of the environment (names, orientation, general knowledge, and spontaneous activity); and the third assesses communication, such as language, interest, reaction to objects, and achievement in communication. The score ranges from 0 to 48, and a lower score indicates a better level of communication (Cronbach α = 0.94). We administered this questionnaire before and after the intervention.

To assess gait and balance, we used the Tinetti scale [29,35], which has two sections. The first evaluates sitting and standing balance and the functions of getting up and sitting using 13 parameters with a maximum score of 16 points (Cronbach α = 0.86). The second evaluates different aspects of walking using nine items with a maximum score of 12 points (Cronbach α = 0.91). The maximum total score is 28 points (Cronbach α = 0.95). We administered this questionnaire before and after the intervention.

As groups were not randomized, we also took into account the variables age at the time of inclusion, sex, and the GDS score [28] in order to assess group comparability. GDS evaluates the degree of cognitive impairment; the initial score is based on the Mini Mental State Examination (MMSE), which considers daily-life activities to assess the individual’s level of dementia. The GDS score ranges from 1 (not impaired) to 7 (very severe cognitive impairment). The GDS stage was assessed by the nursing home psychologist before the intervention.

### 2.5. Statistical Analysis

We expressed quantitative variables as the median and interquartile range (IQR), and qualitative variables as absolute frequencies and percentages. We evaluated group heterogeneity using the Mann–Whitney test for quantitative variables, and the Chi-squared test for qualitative variables. We assessed intragroup differences (postintervention vs. baseline) in the outcome variables using a paired Mann–Whitney test. Finally, to determine the efficacy of the intervention, we performed multivariate linear regression models using the postintervention scores as the response, the group as the predictor and controlling for the respective baseline scores. We also included the interaction between group and baseline scores. All the variables were mean-centered. We carried out all statistical analyses using R software [36], and the level of significance was set to 5% (α = 0.05).

### 2.6. Ethical Considerations

Participation in the study was voluntary, the participants were informed before starting, and were able to leave the study at any time. We treated patients’ personal data confidentially and we used the data solely for this research, in line with the Spanish Organic Law 15/99 on Personal Data Protection (LOPD). This study was certified by the Clinical Research Ethics Committee (CEIC) of IDIAP Jordi Gol (code P17/011). We adhered to a protocol on animal welfare and zoonosis prevention. The study was covered by civil liability insurance for dogs working in therapy.

## 3. Results

### 3.1. Basal Scores

Concerning preintervention values, we found no significant differences for any of the scores measured except for the Tinetti balance variable; the value detected in the EG (median 8.00 (IQR 5.50; 10.0)) was slightly but significantly lower than that in the CG (median 10.0 (IQR 9.00; 11.0); *p* = 0.041). When analyzing the values according to the degree of cognitive impairment, we found no differences between the participants with mild–moderate cognitive impairment (GDS 2–4) and those with severe cognitive impairment (GDS 5–6) (Table 1).

### 3.2. Evaluation of the Intervention

After the intervention, there was a statistically significant improvement in all response variables in both the CG and the EG. The median change in the Holden scale was −1.00 for the CG ((IQR −2.00; 0.00), *p* = 0.004) and −3.00 for the EG ((IQR −5.00; −2.00), *p* < 0.001). The median difference in the Tinetti scale was 1.00 for the CG ((IQR 1.00; 4.00), *p* < 0.001) and 3.00 for the EG ((IQR 2.00; 3.50), *p* < 0.001). The two subdimensions of the Tinetti scale, gait and balance, also showed statistically significant differences after the intervention in both groups.

When stratifying by GDS, we observed similar results than with all participants except from in the Holden scale and the Tinetti gait subdimension. In the case of the Holden scale in patients with mild–moderate (GDS 2–4) cognitive impairment, the EG showed a significant change after intervention (*p* = 0.009), but the CG did not (*p* = 0.098). Regarding the Tinetti gait subdimension, the change after the intervention was not statistically significant in the CG with mild–moderate (GDS 2–4; *p* = 0.174) or severe (GDS 5–6; *p* = 0.149) cognitive impairment. In the EG, there was a significant improvement in the Tinetti gait subdimension in individuals with mild-moderate cognitive impairment (GDS 2–4; *p* = 0.026), but not in those with severe cognitive impairment (GDS 5–6; *p* = 0.065; Table 2).

When comparing the two groups, the improvement in postintervention Holden scale scores (−2.89 points; 95% confidence interval (CI 95% −4.11, −1.67); *p* < 0.001) was significantly better in the EG than in the control group (Figure 1, Table 3). No statistically significant differences were found in any of the Tinetti scale scores, and no differences were found in the effect of the intervention depending on baseline values (Table 3). The results were similar in participants with mild–moderate cognitive impairment (GDS 2–4) and those with severe cognitive impairment (GDS 5–6) (Table 3).

## 4. Discussion

The objective of our study was to investigate the effect of an AAT program on communication and mobility in an institutionalized geriatric population with mild-moderate and severe cognitive impairment or dementia.

AAT has been reported to slow the progression of cognitive decline in patients with mild- or early-stage dementia [37,38], and is considered to offer a social interaction that does not depend on the cognitive level of the recipient [39]. For this reason, patients with mild and moderate cognitive impairment were also included in our study, although previous reports have focused only individuals with severe cognitive impairment or dementia [11,40,41,42,43,44,45,46,47].

In our study, the programmed intervention in people with cognitive impairment was effective in improving mobility (balance and gait, evaluated using the Tinetti scale). The intervention also improved communication, including aspects such as initiative, liking, humor, and interest (evaluated using the Holden scale). Furthermore, our AAT showed a significant improvement in communication skills of almost three points on the Holden scale with respect to the CG.

A limited number of studies have focused on evaluating the effects of AAT on gait and balance in people with mild–moderate and severe cognitive impairment or dementia. The results obtained in the clinical trial performed by Olsen et al. [48] showed a significant effect on balance and the fall risk compared to the control group. Although our exercises were similar to those of Olsen et al. [48], we did not find significant differences between the EG and CG, which may be because in our intervention, (i) the same exercises were also performed in the CG, or (ii) our sample was smaller than that of Olsen et al. Positive effects in the area of physical health had also been studied by Cherniack and Cherniack [47], but with modest results, highlighting the health benefits for elderly people of walking a dog.

Regarding the observed improvement in communication, our results reinforce the idea that group AAT may promote more spontaneous communication and greater interaction between participants. These results are in agreement with those of other studies in the institutionalized geriatric population that suggested that AAT may increase positive social behaviors [26,49] and reduce social isolation and loneliness [15,25]. Similarly, our results were in line with those of Hediger et al. [50], who observed a significant improvement in social behavior during AAT sessions. Swall et al. [51] reported that AAT participants talked more about memories and feelings when the animal was present.

Thus, and in line with previous reports [52], the participation of therapy dogs showed positive results in healthcare settings dealing with people with dementia. The therapies contributed to spontaneity in the sessions and may have helped generate a more relaxed environment that favored communication and improved participants’ ability to concentrate during the session. These inherent features of AAT may help motivate the patient during the treatment process [53,54]. A preliminary study conducted by Menna et al. [55] found that the repeated multisensory stimuli (verbal, visual, tactile) provided by the therapy dog were effective. AAT has been considered a meaningful activity that provides stimulation and social interaction [56,57], and improves the participants’ mood [20,58,59]. Furthermore, interventions that favor communication and an optimistic atmosphere can have a positive impact on social interactions [9,60,61], as our study reflected. These results were in agreement with previous reports indicating that AAT can improve social interactions [11,12] and have a positive effect on communication [62,63], both verbal and nonverbal [64,65].

## 5. Limitations

One of the limitations of our study was the final sample size, which limited our statistical power, although it was indeed bigger than those of other studies [66,67,68]. Increasing the sample size may allow us to observe a significant effect, not only for the communication variable, but also for gait and balance. However, this sample size granted enough statistical power to detect a significant and clinically relevant improvement in the main outcome variables. Another limitation was the open design of the study, in which double-blinding was not possible due to the nature of the intervention. To minimize this limitation, we decided to restrict the EG to one center and the CG to the other, instead of randomizing the entire sample. This lack of randomization is itself another limitation. However, both nursing homes pertain to the same entity, and interventions in both groups (EG and CG) were performed by exactly the same professionals in both centers, apart from the dog. Another limitation was the fragility of the elderly participants, some of whom were at risk of worsening health, which made it difficult for them to participate in some of the sessions. Furthermore, people with dementia had difficulty following the instructions; to mitigate this, sessions were held in small groups.

## 6. Conclusions

This study supports the benefits of AAT in improving communication in people with cognitive impairment in nursing homes. We also observed an improvement in gait and balance in both therapy groups compared to the beginning of the sessions. This demonstrates that AAT might be useful as a complementary, effective treatment for patients with different degrees of cognitive decline. Our results contribute to the scientific basis for using nonpharmacological methods in the care of institutionalized elderly people. Further studies should be performed in order to evaluate the cost-effectivity of the AAT in the treatment of elderly people with cognitive impairment.

## Figures and Tables

**Figure 1 ijerph-17-05899-f001:**
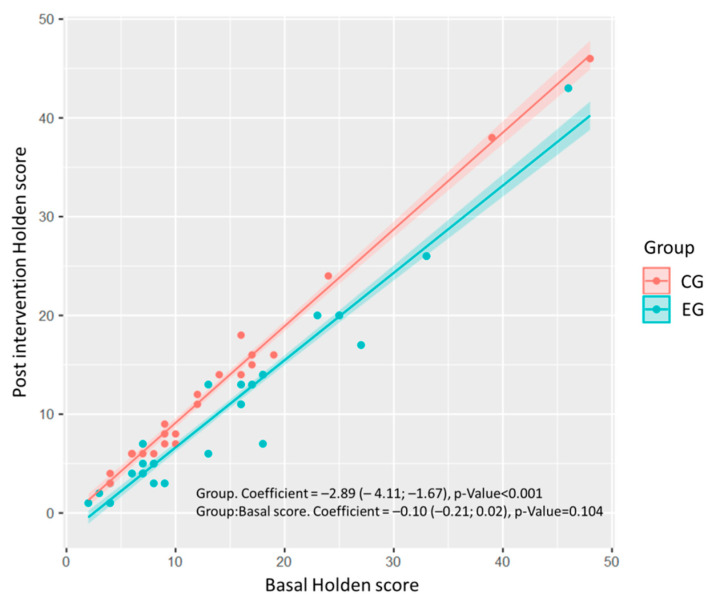
Linear regression model for the response variable “postintervention Holden score” as a function of the group, basal score, and their interaction. Each point represents the participant observed score, and the lines represent the fitted model and 95% confidence interval. The regression coefficients for the group effect are shown, with their corresponding 95% confidence interval and *p*-value. CG: Control group, EG: Experimental group.

**Table 1 ijerph-17-05899-t001:** Baseline demographic characteristics and clinical (preintervention) variables.

	ALL PARTICIPANTS					GDS 2–4					GDS 5–6				
Variables	Total	Control	Experimental	Statistic	*p*-value	Total	Control	Experimental	Statistic	*p*-value	Total	Control	Experimental	Statistic	*p*-value
	N = 46	N = 23	N = 23			N = 15	N = 6	N = 9			N = 31	N = 17	N = 14		
Age (years), *mean* (IQR)	85.0 (80.2; 87.0)	86.0 (80.5; 87.5)	83.0 (80.5; 86.5)	296	0.488	86.0 (80.5; 90.5)	90.0 (87.0; 92.2)	81.0 (80.0; 86.0)	42	0.076	83.0 (80.5; 86.5)	84.0 (80.0; 86.0)	83.0 (81.2; 86.8)	108.5	0.676
Gender, N (%)				0.48	0.489				0.51	0.604				0.52	0.671
Women	35 (76.1)	19 (82.6)	16 (69.6)			11 (73.3)	5 (83.3)	6 (66.7)			24 (77.4)	14 (82.4)	10 (71.4)		
Men	11 (23.9)	4 (17.4)	7 (30.4)			4 (26.7)	1 (16.7)	3 (33.3)			7 (22.6)	3 (17.6)	4 (28.6)		
GDS, N (%)				2.02	0.762				0.63	0.82				0.53	0.707
2	2 (4.35)	1 (4.35)	3 (13.0)			4 (26.7)	1 (16.7)	3 (33.3)			-	-	-		
3	6 (13.0)	3 (13.0)	3 (13.0)			6 (40.0)	3 (50.0)	3 (33.3)			-	-	-		
4	5 (10.9)	2 (8.70)	3 (13.0)			5 (33.3)	2 (33.3)	3 (33.3)			-	-	-		
5	11 (23.9)	7 (30.4)	4 (17.4)			-	-	-			11 (35.5)	7 (41.2)	4 (28.6)		
6	20 (43.5)	10 (43.5)	10 (43.5)			-	-	-			20 (64.5)	10 (58.8)	10 (71.4)		
Outcome variables, *median* (IQR)															
Holden scale	11.0 (7.00; 17.0)	10.0 (7.50; 16.5)	13.0 (7.00; 18.0)	264.5	1	7.00 (4.00; 9.00)	6.50 (4.50; 8.50)	7.00 (4.00; 9.00)	26.5	0.953	16.0 (9.00; 21.0)	14.0 (9.00; 17.0)	16.5 (9.25; 24.5)	103.5	0.538
Tinetti scale															
Total score	14.5 (11.0; 18.8)	16.0 (13.0; 20.0)	13.0 (9.00; 17.0)	353	0.051	13.0 (10.5; 14.5)	14.5 (13.2; 18.8)	11.0 (8.00; 14.0)	44	0.044 *	16.0 (11.0; 19.0)	16.0 (13.0; 20.0)	15.0 (11.0; 18.0)	140.5	0.392
Gait score	5.50 (3.00; 8.00)	6.00 (4.00; 9.00)	5.00 (3.00; 7.00)	334.5	0.122	5.00 (3.00; 6.50)	5.50 (4.25; 7.50)	3.00 (3.00; 6.00)	40	0.123	6.00 (3.50; 9.00)	7.00 (4.00; 9.00)	5.50 (3.25; 8.50)	138	0.448
Balance score	9.00 (7.00; 10.0)	10.0 (9.00; 11.0)	8.00 (5.50; 10.0)	356.5	0.041 *	9.00 (6.50; 9.00)	9.00 (9.00; 11.2)	7.00 (5.00; 8.00)	48	0.012 *	10.0 (7.50; 10.5)	10.0 (8.00; 11.0)	9.00 (7.00; 10.0)	138.5	0.433

Differences were assessed using the Mann–Whitney test for quantitative variables and the Chi-squared test for qualitative variables and by calculating the corresponding *p*-value. *: *p*-value < 0.05; GDS: Global Deterioration Scale of Reisberg. IQR: interquartile range.

**Table 2 ijerph-17-05899-t002:** Comparison between the Holden and Tinetti scale scores of the study participants pre- and postintervention.

TOTAL										
Outcome variables	Control (N = 23)					Experimental (N = 23)				
	Preintervention, median (IQR)	Postintervention, median (IQR)	Difference	U statistic	*p*-value	Pre-intervention, median (IQR)	Postintervention, median (IQR)	Difference	U statistic	*p*-value
Holden scale	10.0 (7.50; 16.5)	9.00 (6.00; 15.5)	−1.00 (−2.00; 0.00)	10	0.004 *	13.0 (7.00; 18.0)	7.00 (3.50; 13.5)	−3.00 (−5.00; −2.00)	0	<0.001 *
Tinetti scale										
Total score	16.0 (13.0; 20.0)	18.0 (14.0; 22.0)	1.00 (1.00; 4.00)	171	<0.001 *	13.0 (9.00; 17.0)	17.0 (11.5; 19.0)	3.00 (2.00; 3.50)	231	<0.001 *
Gait score	6.00 (4.00; 9.00)	6.00 (4.50; 9.00)	0.00 (0.00; 0.50)	21	0.026 *	5.00 (3.00; 7.00)	6.00 (3.50; 7.50)	1.00 (0.00; 1.00)	85.5	0.004 *
Balance score	10.0 (9.00; 11.0)	11.0 (10.0; 13.0)	1.00 (1.00; 3.50)	171	<0.001 *	8.00 (5.50; 10.0)	10.0 (8.00; 12.0)	2.00 (1.00; 3.00)	253	<0.001 *
GDS 2–4										
	Control (N = 6)					Experimental (N = 9)				
Holden scale	6.5 (4.5; 8.5)	6 (4.5; 6.75)	−1 (−1.75; −0.25)	0	0.098	7 (4; 9)	4 (2; 5)	−3 (−6; −2)	0	0.009 *
Tinetti scale										
Total score	14.5 (13.25; 18.75)	19 (16.5; 23.75)	4 (1.75; 4.75)	21	0.035 *	11 (8; 14)	16 (10; 18)	3 (2; 5)	45	0.009 *
Gait score	5.5 (4.25; 7.5)	6 (6; 8.25)	0.5 (0; 1)	6	0.174	3 (3; 6)	5 (4; 7)	1 (0; 1)	21	0.026 *
Balance score	9 (9; 11.25)	13 (10.5; 15.5)	3.5 (1.5; 4)	21	0.035 *	7 (5; 8)	10 (6; 11)	3 (1; 3)	45	0.009 *
GDS 5–6										
	Control (N = 17)					Experimental (N = 14)				
Holden scale	14 (9; 17)	14 (8; 16)	−1 (−2; 0)	7.5	0.023 *	16.5 (9.25; 24.5)	13 (8; 19.25)	−3.5 (−5; −2.25)	0	0.002 *
Tinetti scale										
Total score	16 (13; 20)	17 (14; 21)	1 (0; 3)	78	0.002 *	15 (11; 18)	18 (14; 19.75)	3 (2; 3)	78	0.002 *
Gait score	7 (4; 9)	7 (4; 9)	0 (0; 0)	6	0.149	5.5 (3.25; 8.5)	6 (3.5; 8)	0 (0; 1)	25	0.065
Balance score	10 (8; 11)	11 (10; 12)	1 (0; 3)	78	0.002 *	9 (7; 10)	12 (9.25; 12)	2 (1; 3)	91	0.001 *

The effect observed in the CG and EG was observed at the end of the trial (postintervention) with respect to the corresponding value at baseline (preintervention). The difference was assessed by using a Mann–Whitney test for paired samples. *: *p*-value < 0.05.

**Table 3 ijerph-17-05899-t003:** Evaluation of postintervention scores on the experimental group controlling for the respective basal scores.

TOTAL						
	β_EG_ (95% CI)	*t*-Value	*p*-Value	β_EG:bl_ (95% CI)	*t*-Value	*p*-Value
Holden scale	−2.889 (−4.114–1.665)	−4.761	<0.001 *	−0.096 (−0.212–0.02)	−1.663	0.104
Tinetti scale						
Total score	0.199 (−0.942–1.34)	0.352	0.726	0.08 (−0.121–0.281)	0.802	0.427
Gait score	0.263 (−0.159–0.685)	1.258	0.215	−0.014 (−0.159–0.131)	−0.199	0.843
Balance score	−0.082 (−1.04–0.877)	−0.172	0.864	0.098 (−0.204–0.4)	0.655	0.516
GDS 2–4						
Holden scale	−2.405 (−3.447–1.363)	−5.081	<0.001*	−0.229 (−0.598–0.14)	−1.366	0.199
Tinetti scale						
Total score	0.208 (−2.778–3.194)	0.153	0.881	0.009 (−0.6–0.618)	0.032	0.975
Gait score	−0.018 (−1.02–0.985)	−0.039	0.97	0.18 (−0.255–0.615)	0.91	0.382
Balance score	0.597 (−2.139–3.334)	0.481	0.64	−0.487 (−1.472–0.498)	−1.088	0.3
GDS 5–6						
Holden scale	−2.688 (−4.089–1.288)	−3.939	<0.001 *	−0.087 (−0.213–0.039)	−1.412	0.169
Tinetti scale						
Total score	0.475 (−0.752–1.701)	0.795	0.434	0.056 (−0.159–0.27)	0.532	0.599
Gait score	0.299 (−0.191–0.789)	1.252	0.221	−0.05 (−0.209–0.109)	−0.647	0.523
Balance score	0.207 (−0.894–1.307)	0.386	0.703	0.106 (−0.266–0.478)	0.585	0.564

Linear regression coefficients of the experimental group (βEG) and experimental group–baseline score interaction (βEG:bl) on postintervention scores, with their respective 95% confidence intervals (95% CI) and *p*-values. *: *p*-value < 0.05.

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
