# Peer review of "Animal-Assisted Therapy Improves Communication and Mobility among Institutionalized People with Cognitive Impairment"

_ijerph, 2020, doi:10.3390/ijerph17165899_

Round 1

Reviewer 1 Report

The reviewer doesn't think the explanation is sufficient throughout. For example, center 1 was the AAT group and center 2 was the control group, but there are not enough comments on it.

There is no explanation of what the dog is doing. Is the dog simply there?

The reviewer can't find any comments about the burden on the dog. There are 6 participants in one group, probably 4 groups, and one dog for AAT that lasts 60 minutes of sessions for 12 weeks.

The role of dogs in this program is unknown and referees do not have an overview of AAT. A detailed explanation is needed.

Animals are essential for AAT, but there are no comments in the intro.

Reviewer 2 Report

Dear Authors,

I think that by this article you have addressed a very interesting topic and currently little covered in the literature. Unfortunately, I believe that your paper is not currently adequate for publication because it needs profound and substantial changes.

Reviewer 3 Report

Overall, this paper represents a significant addition to the field and I believe would be an acceptable manuscript for IJERPH given the following revisions below. Both the introduction and discussion of the paper are well-written, concise, and appropriately cited. The methods need improvement to be able to fully replicable from an independent research group. The findings provide important preliminary results suggesting a social communication benefit of animal-assisted therapy as an adjunct to physiotherapy for individuals with cognitive impairment. A major design weakness is the assignment of experimental and control groups based on residential center, rather than across center. This may largely limit the interpretation of the findings due to unexplained across-center and across-therapist variance, which the authors do not address as a limitation in the discussion. In summary, I recommend this manuscript be revised and resubmitted, with the following comments in mind:

Introduction:

Even as an AAI researcher myself, I believe it is critically important to have clear communication regarding the very preliminary and limited nature of this field. For example, on lines 65-67 the text should read “AAT sessions may have a relaxing effect on people with dementia” rather than to mislead readers that this is something proven or widely known. It is also important to state findings in relation to their population, since our field is growing and preliminary. Lines 67-68 should ready “One study found that patients with brain injuries receiving AAT focused on the same activity for a longer time and demonstrated greater enthusiasm for new exercises” Line 69 should read “Regarding mobility, there is some emerging evidence that AAT may benefit patients with hemiparesis due to stroke 69 [17-19], with aphasia [20], and with traumatic brain injury [21].” (Note that references 19,20, and 21 were preliminary pilot studies)

Methods:

Please provide a detailed description of how voluntary informed consent or assent was obtained from patients, especially those with moderate to severe cognitive impairment. How was comprehension assured?

Please state the exclusion reasons for the N=29 excluded patients (i.e., how many were excluded for age, severe cognitive deterioration, fear of dogs, etc.) either in the text or the flow diagram. The information on the number of individuals with dog allergies or fear of dogs is especially helpful for future researchers in planning sample sizes/designs in this population.

Please clarify if the two primary care nurses facilitating sessions were identical across both nursing homes (two nurses for the whole study) or if two primary care nurses facilitated sessions within each nursing home (four nurses for the whole study).

The statement in line 135 regarding the therapy dog’s monitoring is unclear. Was his trainer/handler directly responsible for the welfare and stress behavior monitoring of the dog for all sessions? Also, authors state that patients received one 60-min session per week for 12 weeks, but were not clear on the number of sessions the therapy dog conducted (I assume multiple sessions per week?). As a final consideration of the dog’s welfare, line 173 says “a protocol was followed” but I suggest specifying which protocol and/or clarifying if any external review was done.

Please provide the internal reliability of each measure (Cronbach's α).

More detail is required for the Global Deterioration Scale in terms of when the assessment was given, who it was given by, and its stability over the experimental period.

Results

Control variables: Can you explain what is meant by “considered”? Were and sex an explanatory variable in any outcomes? It is unclear which models controlled for age/sex, and which didn’t.

It is not clear why linear regressions were performed in the results, which wasn’t stated in the analysis section of the methods. It was also not clear where ANCOVA results were presented. Clarification is needed in the tables/results by specifying which analyses were performed in each table (e.g., “Table 2: Comparison between the Holden and Tinetti scale scores of the study participants pre- and post-intervention using ANCOVA controlling for X,Y,Z”) as well as reporting of U and F values where appropriate.

Discussion

As mentioned in feedback regarding the introduction, I would urge authors to use language suggestive of the preliminary nature of both the study and this field (“may” “some evidence for” “one study found”, stating population). In lines 270-272, it is important to not overstate the findings – these mechanisms may be at play, but were not measured. In addition, I am uncomfortable with the subjective observations regarding participant’s calmness and positive mood in the experimental condition being mentioned in lines 275-277, as these outcomes were not systematically assessed and are only speculation.

A critical finding of this paper is that AAT in addition to physiotherapy improved communication and gait only in individuals with mild or moderate impairment, while those with severe impairment did not differ from controls. I was surprised that this finding was not mentioned in the abstract or the conclusion.

The authors do not address the critical limitation of unexplained across-center and across-therapist variance. In particular, it is unknown if the experimental group nursing home simply created an environment in which more communication and trust were expressed compared to the control group nursing home. This limitation is severe and warrants a discussion of implications to the internal and external validity of the findings.

In the conclusion, I would urge authors to not overstate their findings. Specifically, it should be mentioned that both experimental groups and control groups significantly benefited (a very important and positive finding) And, the addition of a therapy dog resulted in more communication improvement compared to the control group, but only among those with mild-moderate impairment. I would not confidently say that this finding encourages the widespread use of therapy dogs in physiotherapy sessions as argued, but rather, the inclusion of a therapy does not seem to have any negative effects and may result in some positive social outcomes.

Round 2

Reviewer 1 Report

No comments for Authors.

Reviewer 2 Report

The authors have made the right changes. The paper can be accept in the present form